# Effectiveness Evaluation of Repetitive Transcranial Magnetic Stimulation Therapy Combined with Mindfulness-Based Stress Reduction for People with Post-Stroke Depression: A Randomized Controlled Trial

**DOI:** 10.3390/ijerph20020930

**Published:** 2023-01-04

**Authors:** Haoran Duan, Xin Yan, Shifeng Meng, Lixia Qiu, Jiayu Zhang, Chunxia Yang, Sha Liu

**Affiliations:** 1Nursing College, Shanxi Medical University, Taiyuan 030001, China; 2Drug Clinical Trial Institution, First Hospital of Shanxi Medical University, Taiyuan 030001, China; 3Rehabilitation Medicine Centre, First Hospital of Shanxi Medical University, Taiyuan 030001, China; 4Epidemiology and Health Statistics, College of Public Health, Shanxi Medical University, Taiyuan 030001, China; 5Mental Health and Psychiatry, First Hospital of Shanxi Medical University, Taiyuan 030001, China

**Keywords:** mindfulness-based intervention, rehabilitation nursing, post-stroke depression, repetitive transcranial magnetic stimulation, depression, cognitive function

## Abstract

Background: Post-stroke depression (PSD) is most prevalent during the rehabilitative period following a stroke. Recent studies verified the effects of repetitive transcranial magnetic stimulation therapy (rTMS) and mindfulness-based stress reduction (MBSR) in patients with depression. However, the effectiveness and prospect of application in PSD patients remain unclear. This study sought to evaluate the effectiveness of a combined intervention based on rTMS and MBSR for the physical and mental state of PSD patients. Methods: A randomized, double-blind, sham-controlled study design was employed. Participants were recruited from the Rehabilitation Medicine Centre and randomly assigned to receive either MBSR combined with active or sham rTMS or sham rTMS combined with general psychological care. We used a 17-item Hamilton Depression Rating Scale (HAMD-17), a mini-mental state examination (MMSE), the Modified Barthel Index (MBI), and the Pittsburgh Sleep Quality Index (PSQI) to evaluate depressed symptoms, cognitive function, activities of daily living (ADL), and sleep quality at baseline, post-intervention, and the 8-week follow-up. A two-factor analysis of variance was used to compare differences between groups, and Pearson’s linear correlation was used to analyze the possible relationship between variables and potential predictors of depression improvement. Results: Seventy-two participants were randomized to rTMS–MBSR (*n* = 24), sham rTMS–MBSR (*n* = 24), or sham rTMS–general psychological care (*n* = 24). A total of 71 patients completed the questionnaire, a 99% response rate. There were significant time and group interaction effects in HAMD-17, MMSE, MBI, and PSQI scores (*p* < 0.001). The repeated-measure ANOVA showed a significant improvement of all variables in rTMS–MBSR compared to sham rTMS–MBSR and sham rTMS combined with general psychological care (*p* < 0.05). Additional results demonstrated that cognitive function, sleep quality, and activities of daily living are associated with depressive symptoms, and cognitive function is a potential variable for improved depression. Conclusion: Depressive symptoms can be identified early by assessing cognitive function, and rTMS–MBSR might be considered a potentially helpful treatment for PSD.

## 1. Introduction

Stroke was the third highest cause of death and disability (5.7% [5.2–6.2] of total disability-adjusted life years (DALYs)) and the second highest cause of death globally (11.6% [10.8–12.2] of total deaths) in 2019 [1]. Post-stroke depression (PSD) is the most prevalent neuropsychiatric disorder following stroke, with a cumulative incidence of 52% within 5 years [2]. Robinson et al. [3] discovered that the severity of impairments in activities of daily living, social functioning, and cognitive function were all associated with the occurrence of PSD in the first systematic longitudinal study. In addition, PSD can severely influence patient quality of life and rehabilitation [4,5,6]. These issues adversely impact disease prognosis and medical costs [7].

Anti-depressants are still mainstream treatments for PSD. However, many patients do not improve with such therapy because of the long treatment cycle and adverse reactions [8]; therefore, clinical researchers try to test effective complementary therapies for PSD. Repetitive transcranial magnetic stimulation (rTMS) is a non-invasive neurophysiological technique that has been widely employed in the field of neurorehabilitation [9] and is safe and effective in treating major depression and facilitating neuroplasticity [10]. Recent studies have verified the effectiveness and potential mechanisms of rTMS in improving cognitive function and depression for PSD [11,12,13]. In addition, the dorsolateral prefrontal cortex (DLPFC), a critical brain region for emotion regulation, is often an essential stimulation target for rTMS [14]. However, some researchers discovered that rTMS raised the risk of headaches, which may be related to the psychological stress caused by wearing the treatment cap for the first time [12,15].

Together with rTMS, one innovative strategy includes mindfulness-based interventions (MBIs) such as mindfulness-based stress reduction (MBSR). Mindfulness is an inner state of consciousness with an operational definition that includes “self-regulation of attention” and “orientation to experience” [16]. MBIs, including MBSR and mindfulness meditation, have been found to increase the ability of depressed individuals to inhibit negative stimuli by increasing DLPFC activation levels [17,18,19]. A modest number of studies have also reported mindfulness as a promising method for improving stroke recovery and enhancing post-stroke patients’ limb function and level of activity [20,21,22]. However, MBSR has not been widely studied with regard to improving PSD clinical outcomes.

Based on this evidence, our study team hypothesized that rTMS combined with MBSR could result in greater synergistic effects than independently using either therapy. To our knowledge, no clinical trial investigating the efficacy of rTMS–MBSR for PSD patients has been published. Our study used a Modified MBSR Curriculum introduced by Gray (a stroke survivor and certified MBSR instructor) to guide clinical operation [23]. To clarify whether rTMS–MBSR should be considered an effective alternative therapy, our aim was to analyze depressive level, cognitive function, activities of daily living, and sleep quality to evaluate the effects of rTMS–MBSR for PSD patients in the rehabilitation period. Furthermore, we explored the possible correlations between cognitive function, activities of daily living, and sleep quality, and depression. The results will guide clinical practice and inform deeper studies using electroencephalography (EEG).

## 2. Materials and Methods

### 2.1. Study Design

Recruitment occurred at the rehabilitation medical center of Shanxi Medical University’s First Hospital between 2 January 2022 and 28 August 2022. Potential participants who were screened by clinical research physicians were scheduled for an on-site consent meeting with a clinical researcher. Participants received no financial compensation in this study. Figure 1 shows the CONSORT flow diagram.

We designed a randomized, double-blind (blinded participants and raters), sham-controlled research trial to determine the efficacy of rTMS stimulation in conjunction with MBSR. The assignment of participants to treatment and sham/control groups was random. Randomization was performed using Visual Binning in IBM SPSS Statistics v.26 software:Group A: real rTMS stimulation and MBSR (rTMS–MBSR);Group B: sham rTMS stimulation and MBSR (sham rTMS–MBSR);Group C: sham rTMS stimulation and general psychological care.

This experiment was conducted on 72 patients with mild to moderate post-stroke depression admitted at the medical rehabilitation center of Shanxi Medical University’s First Hospital in China. One patient in group A dropped out of the study due to unwillingness to cooperate with the questionnaire. A total of 71 participants were included in the study. 

### 2.2. Inclusion and Exclusion Criteria

The inclusion criteria for the patients were as follows: (1) diagnosed with depression due to a stroke according to the *Diagnostic and Statistical Manual of Mental Disorder, Fifth Edition* (DSM-5) [24]; (2) aged 18 to 75 years old who were diagnosed with depression nearly 1 month into the rehabilitation stages (1–6 months post-stroke); (3) the imaging changes of cerebral infarction confirmed by cranial CT and MRI; (4) a 17-item Hamilton Depression Rating Scale (HAMD-17) score at admission of 8–24 points; and (5) having read and signed the informed consent form. Patients taking 5-hydroxytryptamine reuptake inhibitors (SSRIs) were permitted to participate if the drug had been taken for at least three months before the commencement of the study and the dosage had remained steady for the preceding 60 days. 

The exclusion criteria for PSD were as follows: (1) HAMD-17 score more than 24 points; (2) diagnosed with other neurological or psychiatric disorders different from PSD; (3) having metal grafts in their bodies; (4) a history of epilepsy, Parkinson’s disease, and brain injury, or an unable to cooperate due to irritability and aggressivity; and (5) aphasia and disturbance of comprehension.

### 2.3. Sample Size Estimation

The study’s primary aim was to compare changes in the HAMD-17 from baseline to post-test and the 8-week follow-up between the 3 groups. We made assumptions for calculating the sample size based on data from a randomized controlled trial that reported significant improvement in depression symptoms [25]. In repeated measures analysis using PASS software, the sample sizes for the three groups whose means were being compared were 24, 24, and 24. The total sample of 72 subjects achieved 90% power to detect differences between the means versus the alternative of equal means using an F test with a 0.05 significance level. The standard deviation of the means, 1.35, represents the magnitude of the variation in the means. It was assumed that the standard deviation within a group was 2.14 [26,27,28].

### 2.4. Intervention

All participants were given routine standard of care medical treatment, including exercise therapy, blood pressure control, blood glucose control, improvement of blood supply to the brain, neurotrophic support, maintenance of acid–base balance, symptomatic interventions, medication nursing, and health education. The combined intervention group, mindfulness intervention group, and general intervention group received rTMS–MBSR, sham rTMS–MBSR, and sham rTMS–general psychological care, respectively, based on these conventional therapies.

#### 2.4.1. Repetitive Transcranial Magnetic Stimulation

A Wuhan OSF-3 Transcranial Magnetic Stimulator was used to achieve rTMS therapy. The stimulation coil was positioned in the motor area of the cerebral cortex’s functional region corresponding to the thumb. We selected the left dorsolateral prefrontal cortex (DLPFC) in this study. Magnetic stimulation frequency was 10 Hz; magnetic stimulation strength was 80% of motor threshold level; each sequence of 4 s continuous stimulation was followed by a 56 s interval; there were 20 sequences (about 20 min duration) and a total of 1400 pulses per day. The sham and control groups utilized a sham coil to imitate auditory and somatic sensations. Each group received 20 rehabilitation sessions over 4 weeks (1 session per day for 5 days per week). Additionally, throughout each treatment, mindfulness guidance audio was played for patients.

#### 2.4.2. Mindfulness-Based Stress Reduction

By consulting each patient’s case, the intervener learned about the patient’s physical and mental condition, completed pre-class communication, gained trust, built confidence in rehabilitation, and established effective contact between the psychological counselor and the patient or caregiver. The intervener taught the first three courses and the pilot course on the second level of the rehabilitation hall. The remaining three lessons were implemented at the Tencent conference. The patient was required to finish the related homework after each class. The assigned task was to follow the audio and spend at least 20 min practicing mindfulness, six times weekly. Patients who participated in fewer than four sessions were considered to have withdrawn from the study.

The MBSR intervention was led by a trained instructor who had obtained an MBSR certificate of completion and had over two years of background experience in MBSR. In addition, the patient’s treatment time (28 days) and physical impairment may also complicate the MBSR curriculum’s execution. This study abolished retreat days and adopted Gray’s Modified MBSR Curriculum, the adaptations of which were developed using the 2017 MBSR Authorized Curriculum Guide [23,29]. The STOP technique in the third week was created by Stahl and Goldstein [30], and the RAIN technique in the fifth week was adopted by Brach [31]. The main contents of the MBSR program were body scan meditation, mindful walking meditation, yoga/mindful movement sequences, and seated mindfulness meditation (see Appendix A). Each group received 6 sessions over 6 weeks (1 session/hour/week). Participants in groups A and B were given workbooks, audio CDs, and instructions for home practice.

#### 2.4.3. General Psychological Care

Clinical Registered Nurses from the Rehabilitation Medicine Centre completed general psychological care for participants: (1) Building positive interpersonal relationships while focusing on patients. (2) Providing patients with stroke-related information to improve their understanding of the condition and to alleviate negative emotions such as anxiety and fear. (3) Motivating patients to assume responsibility for their recovery and daily activities. (4) Encouraging patients to pursue new interests and increase their leisure time. (5) Informing family members of stroke-related information and encouraging them to offer the patient better care and support. General psychological care sessions were timed to match the MBSR sessions in frequency and duration.

### 2.5. Data Collection

General information collection included age, gender, marriage, occupation, educational level, annual family income, lesion location, and lesion hemisphere. All participants were measured before (T_0_) and after the treatment (T_1_) and during the last follow-up after about 8 weeks (T_2_); in particular, they were evaluated through a 17-item Hamilton Depression Rating Scale (HAMD-17), a mini-mental state examination (MMSE), the Pittsburgh Sleep Quality Index (PSQI), and the Modified Barthel Index (MBI). 

To improve the quality of the survey, patient questionnaires were personally assessed by their supervising physician, who could then immediately assist the patient in answering any non-responsive questions. Except for one patient in the combined treatment group, who refused to participate in the questionnaire before the intervention, all patients completed the collection of data at three time points.

### 2.6. Outcome Measures

#### Clinical Assessment

*Co-primary Outcomes* The scores of the HAMD-17 and MMSE were examined. The HAMD-17 is a valid and reliable 17-item scale that rates each item’s severity on a 0–2 or a 0–4 scale for depression [32,33]. A score of 0–7 on the HAMD-17 is typically seen as being within normal bounds (or in clinical remission). For enrollment into a clinical trial, a score of 20 or above often indicates severe depression, and exclusion from the clinical trial was necessary [34]. The MMSE, which measures six different aspects of cognition, including visuospatial, executive function, immediate memory, attention, computation, delayed memory, and orientation, was used to evaluate cognitive function. Greater cognitive function is indicated by higher scores (range 0–30) [35].

*Secondary Outcomes* The Pittsburgh Sleep Quality Index (PSQI) is a self-reporting questionnaire created by researchers at the University of Pittsburgh to assess sleep quality. The PSQI consists of seven sub-components and 19 individual questions. (range 0–21; higher scores indicate greater sleep problems) [36,37]. Activities of daily living were measured with the 10-item Modified Barthel Index (MBI, 0–100 scores; lower scores indicate poor exercise capacity) [38]. 

### 2.7. Statistical Analysis

The statistical analysis employed IBM SPSS Statistics v.26 software. Using a one-factor analysis of variance, the potential difference between the baseline characteristics of the three groups was determined. Using a two-factor, Group (A, B, C) and time, analysis of variance, the statistical differences between pre- and post-treatment (T_0_, T_1_, and T_2_) in terms of rTMS–MBSR were evaluated. For post hoc comparisons, the Bonferroni test was utilized. *p* < 0.05 was used as the statistic significance cutoff level. In addition, the first case of Pearson’s linear correlation was used to assess the possible association between cognitive function, daily life activities, and quality of life, and depression state at each investigated time (T_0_, T_1_, T_2_). Similarly, correlation analysis was used to assess the predictive value of the cognitive function by connecting its values at T_0_ with the depression established at T_2_ for all patients. 

### 2.8. Ethics Statement

The First Hospital of Shanxi Medical University’s institutional review board gave its approval for this study, which was carried out in accordance with the Declaration of Helsinki ([2022] (Y2)). The protocol of this trial was registered in the Chinese Clinical Trial Registry (ChiCTR2200061523). All participants in the study provided their informed consent.

## 3. Results

The demographic characteristics of the participants are shown in Table 1. No significant differences were observed in the baseline values between groups A, B, and C for all parameters.

### 3.1. Depressive State

A two-factor repeated measures ANOVA found a significant time main effect (*F* = 183.610, *p* < 0.001), interaction effect between group and time (*F* = 11.157, *p* < 0.001), and group main effect (*F* = 17.997, *p* = 0.001) for the three groups. Post hoc comparison of the simple effect of time showed that HAMD-17 scores of the three groups decreased after intervention and the 8-week follow-up (*p* < 0.05) compared to pre-intervention. After intervention and the 8-week of follow-up, the HAMD-17 score of group A < group B < group C (*p* < 0.05). At the 8-week follow-up, a notable +0.16 worsening of the HAMD-17 score was found in the control group (Table 2).

### 3.2. Cognitive Function

A two-factor repeated measures ANOVA found a significant time main effect (*F* = 63.639, *p* < 0.001), interaction effect between group and time (*F* = 5.114, *p* = 0.001), and group main effect (*F* = 7.594, *p* = 0.001) for the three groups. Post hoc comparison of the simple effect of time showed that MMSE scores of the three groups increased after intervention and the 8-week follow-up compared to pre-intervention (*p* < 0.05). After intervention and the 8-week follow-up, the MMSE score of group A > group B > group C (*p* < 0.05). At the 8-week follow-up, a decline in MMSE scores was observed in groups B and C (Table 3).

### 3.3. Activities of Daily Living

A two-factor repeated measures ANOVA found a significant time main effect (*F* = 400.377, *p* < 0.001), group and time interaction effect (*F* = 2.903, *p* = 0.024), and group main effect (*F* = 3.297, *p* = 0.043) for the three groups. Post hoc comparison of the simple effects of time found that ADL scores were higher in all three groups after the intervention and at the 8-week follow-up compared to pre-intervention (*p* < 0.05). After intervention and at the 8-week follow-up, ADL scores were higher in group A than in groups B and C (*p* < 0.05) (Table 4).

### 3.4. Sleep Quality

A two-factor repeated measures ANOVA found a significant time main effect (*F* = 192.487, *p* < 0.001), group and time interaction effect (*F* = 19.629, *p* < 0.001), and group main effect (*F* = 5.603, *p* = 0.006) for the three groups. Post hoc comparison of the simple effect of time found that PSQI scores were lower in all three groups after the intervention and the 8-week follow-up compared to pre-intervention (*p* < 0.05). PSQI scores in group A < group B after the intervention (*p* < 0.05); there was no significant difference between group B and C (*p* > 0.05), whereas PSQI scores in group A < group B < group C at the 8-week follow-up (*p* < 0.05) (Table 5).

### 3.5. Correlation and Predictive Role of Cognitive Function, Activities of Daily Living, and Sleep Quality with Depression

To investigate the potential relationship between variables, a Pearson’s correlation analysis was performed on all individuals at each time point (T_0_, T_1_, T_2_) between the MMSE, MBI, PSQI, and HAMD-17 scores. The results showed that the HAMD-17 scores negatively correlated with MMSE and MBI scores (r = −0.455, *p* < 0.001; r = −0.580, *p* < 0.001). Furthermore, there was a positive correlation between HAMD-17 and PSQI scores (r = 0.410, *p* < 0.001) (Figure 2), underlying the higher MMSE and MBI scores, or the lower PSQI scores, the lower HAMD-17 scores, which corresponds to a better psychological state.

In addition, correlation analyses were also performed on all subjects as a group at baseline to explore the possible predictive role of each variables. The results showed that HAMD-17 at the 8-week follow-up (T_2_) correlates negatively with MMSE (T_0_) (r = −0.263, *p* = 0.027) (Figure 3), highlighting the possibility of forecasting the improvement of the depression state by analyzing cognitive function at the beginning of the training session.

## 4. Discussion

Post-stroke depression, one of the most prevalent clinical complications of a stroke, poses a formidable obstacle for stroke survivors during rehabilitation. The primary objective of this study was to evaluate the synergistic effects of 4-week rTMS combined with 6-week MBSR for PSD patients. After the experimental protocol and throughout the 8-week follow-up, patients who received the combined treatment demonstrated significant improvements in depressed mood, cognitive function, activities of daily living, and sleep quality. In addition, there is tentative evidence of a possible association between cognitive function, activities of daily living, and sleep quality, and depressed mood. As far as we know, this is the first study to examine the efficacy of rTMS–MBSR therapies for recovering PSD patients.

Regarding the primary objectives of our study, we discovered that following the intensive daily treatment, immediate and follow-up results indicated a significant improvement in depression status in PSD patients treated with rTMS–MBSR. It has been shown in other studies that mindfulness-based stress reduction combined with transcranial magnetic stimulation effectively enhances the comfort degree of psychology [39]. Although the research subjects were patients with generalized anxiety disorder and the heterogeneity of mindfulness-based interventions makes our conclusion somewhat premature, this research field indicates that rTMS and mindfulness training assist PSD patients in enhancing their health-related quality of life [40,41,42,43]. As argued before, Li et al. explained this finding by the fact that rTMS improves brain nerve function in PSD patients [44]. Exploring the neural mechanisms of rTMS–MBSR is also a future research direction for our research team. In addition, Zhang et al. reported that rTMS intervention was no longer statistically significant by the sixth weekend [45]. We modestly suggest that the synergistic effect of MBSR allows the depression state of patients receiving rTMS to remain valid over the 8-week follow-up period.

The significant improvement in cognitive function is in line with that reported by Kim et al. [46] in their retrospective study at the rehabilitation center of a university-affiliated general hospital and Zou et al. [22] in their “A Systematic Review with Meta-Analysis”. They found a high gain in the MMSE total score and sub-scores of all domains, including attention and concentration, registration, and recall. This finding could also be explained by the fact that rTMS improves working memory, which is the core of cognitive function [47]. Notably, MBSR alone showed significant improvements in depression and cognitive function compared with those in group C. This finding is consistent with the Broaden-and-Build Theory and Mindfulness-to-Meaning Theory, which support that mindfulness-based interventions are effective in improving the cognitive functioning of depressed individuals and reducing the negative cognitive bias caused by cognitive deficits [48,49]. This further support that MBSR may be a necessary complementary treatment to help stroke survivors. It is surprising that the rTMS–MBSR group showed a slight improvement during the 8-week follow-up period compared to immediate results. This finding should be investigated further by enlarging the sample size and extending the follow-up time.

Our findings suggest that these processes mediate some of the effects of MBI and PSQI on PSD patients. rTMS and MBSR promote the recovery of ADL and are closely related to the improvement of cognitive function. Cognitive impairment in patients can lead to a decrease in ADL, and early cognitive function can even be used as an independent predictor of long-term functional recovery and ADL [50]. Similar results have supported the therapeutic role of rTMS interventions in reducing cocaine use and accompanying symptoms, such as sleep disturbance [51]. Xue et al. [52] found that the differences in total PSCI score and each factor score of mindfulness training were statistically significant at 4 and 6 weeks of intervention. In our study, patients reported coping with insomnia with body scans.

Finally, the correlation analysis revealed that the HAMD-17 score was strongly negatively correlated with MMSE and MBI scores and positively correlated with PSCI scores. The prognosis of depression symptoms is affected by cognitive function analyzed at the baseline. A cognitive perspective for depression and Monitor and Acceptance Theory (MAT) suggest that depressive symptoms are associated with cognitive impairment and that improved cognitive function can alleviate negative mood and stress-related physical and mental problems [53,54]. A study published in *JAMA Psychiatry* found that an increased connection between the brain’s lateral orbital frontal cortex, cuneus, and dorsolateral prefrontal cortex caused patients to experience negative emotions for an extended period, leading to a reduction in sleep quality [55]. In addition, ADL has been proven to be an independent risk factor for post-stroke depression, and reducing functional impairment should be a rehabilitation goal for patients [56].

A meta-analysis indicated that the risk of headache was significantly higher in the rTMS group compared with the control group (OR, 3.53; 95% CI, 1.85–8.55; *p* < 0.001; I2, 0%; *n* = 496). This may be related to the psychological pressure caused by the initial use of this treatment [12]. However, no adverse reactions such as headaches were reported in our study. Some patients reported relief of tension through short mindfulness exercises during brain stimulation treatment.

## 5. Limitations and Further Perspectives

This study has limitations. The number of patients recruited is constrained, with more male patients than female patients, according to the availability of our medical rehabilitation center. The results of the present study indicate that the use of rTMS has the potential to produce long-lasting benefits. In the future, we would increase the sample size to design a fourth group of rTMS to emphasize the results obtained in the present study. We cannot generalize our conclusion to other stroke patients with aphasia and comprehension impairment because they cannot communicate or comprehend our instructions; future studies must be modified to suit such patients. A significant limitation was that we combined online and offline approaches to ensure the intervention’s dose and fidelity because of the patients’ actual hospitalization cycle. This approach might be biased. However, patients receiving offline treatment may experience greater group support, and some claimed that getting help online was simpler. The outcomes also demonstrated the viability of the online intervention, although further research is required to determine its efficacy.

## 6. Conclusions

PSD is much more common during the post-stroke rehabilitation period than it is at other times, which not only negatively affects patient rehabilitation but also lowers their level of functional independence and raises their mortality risk. Therefore, it is crucial to recognize and treat PSD throughout rehabilitation. Our findings do demonstrate that rTMS–MBSR can successfully enhance patients’ mental health and delay the development of cognitive impairment. Our mindfulness-based stress reduction program is similarly helpful for PSD patients who cannot receive repetitive transcranial magnetic stimulation. A thorough assessment of the patient’s cognitive function can identify the symptoms of depression as early as possible. Given the high completion and attendance rate, we believe that the combined intervention based on rTMS and MBSR can be effectively promoted in rehabilitation medical centers. Additionally, nurses and patient family members should consider opportunities for mindfulness-based intervention training and incorporate these initiatives into practice.

## Figures and Tables

**Figure 1 ijerph-20-00930-f001:**
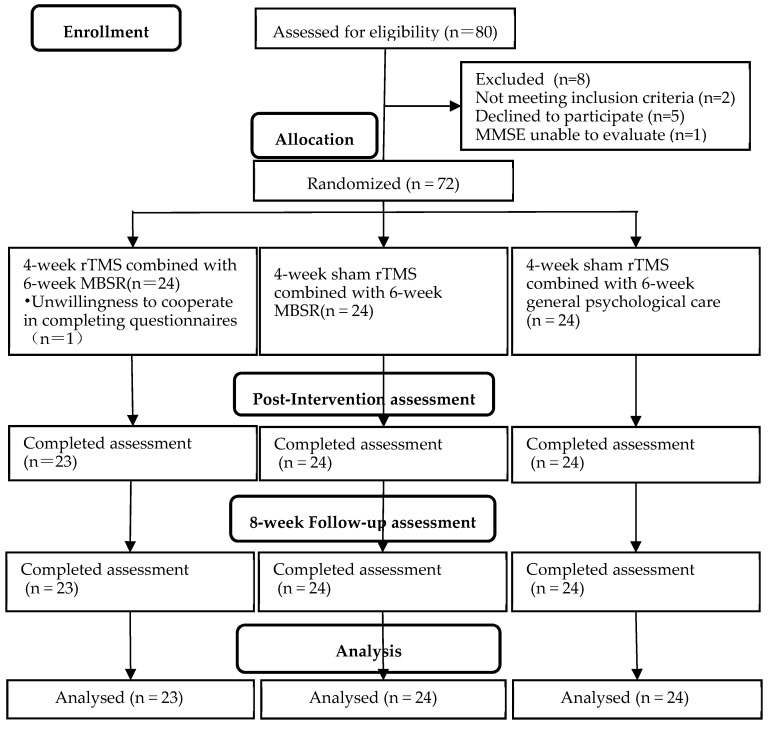
CONSORT flow diagram.

**Figure 2 ijerph-20-00930-f002:**
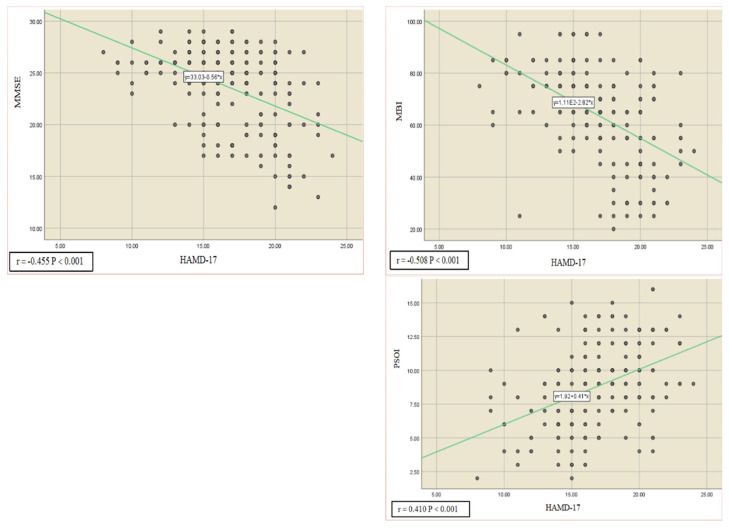
Pearson’s correlation analysis was used to consider all subject in T_0_, T_1_, T_2_ between MMSE, MBI, PSQI, and HAMD-17 scores, showing a strong negative correlation between HAMD-17 and MMSE (r = −0.455, *p* < 0.001), a positive correlation between HAMD-17 and PSQI (r = 0.410, *p* < 0.001), and a negative correlation between HAMD-17 and MBI (r = −0.508, *p* < 0.001). The lower PSQI score and the higher MMSE and MBI score, the lower HAMD-17 score, corresponding to a better psychological state.

**Figure 3 ijerph-20-00930-f003:**
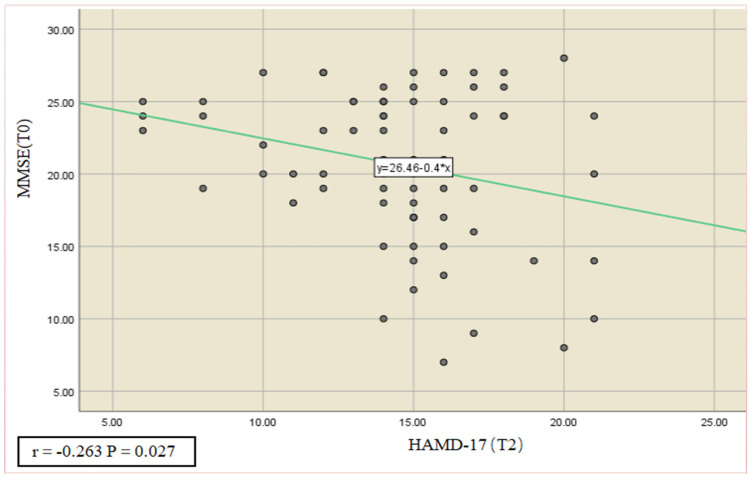
Pearson’s correlation analysis was performed on the cognitive roles of all subjects as a group at T_0_ and HAMD-17 scores at T_2_, showing that HAMD-17 at the 8-week following up correlates negatively with MMSE (r = −0.026, *p* < 0.05).

**Table 1 ijerph-20-00930-t001:** General characteristics of subjects (*n* = 71).

Parameters	Group A (*n* = 23)	Group B (*n* = 24)	Group C (*n* = 24)	χ2/F Value	*p* Value
Gender, *n* (%)				0.158	0.924
Male	19 (82.6)	20 (83.3)	19 (79.2)		
Female	4 (17.4)	4 (16.7)	5 (20.8)		
Age (year)	58.30 ± 13.06	53.63 ± 13.01	54.42 ± 14.27	0.808	0.450
The course of stroke (day)	72.57 ± 20.62	74.00 ± 18.55	70.88 ± 20.73	0.147	0.864
Education, *n* (%)				2.971	0.563
Primary school and below	6 (26.1)	11 (45.8)	8 (33.3)		
Junior high school	5 (21.7)	4 (16.7)	7 (29.2)		
High school and above	12 (52.2)	9 (37.5)	9 (37.5)		
Family annual income, *n* (%)				0.599	0.741
<100,000	13 (56.5)	13 (54.2)	11 (45.8)		
≥100,000	10 (43.5)	11 (45.8)	13 (54.2)		
Marital status, *n* (%)				6.275	0.180
Bereaved spouse	1 (4.3)	0 (0)	2 (8.3)		
Unmarried	0 (0)	0 (0)	2 (8.3)		
Married	22 (95.7)	24 (100)	20 (83.4)		
Occupation, *n* (%)				7.572	0.109
Unemployed	2 (8.7)	10 (41.7)	6 (25)		
Retired	7 (30.4)	5 (20.8)	4 (16.7)		
Employed	14 (60.9)	9 (37.5)	14 (58.3)		
Lesion Location, *n* (%)				6.654	0.354
Brainstem	3 (13)	8 (33.3)	3 (12.5)		
Cerebellar	3 (13)	0 (0)	3 (12.5)		
Cortical–sub-cortical	8 (34.8)	8 (33.3)	9 (37.5)		
Sub-cortical	9 (39.1)	8 (33.3)	9 (37.5)		
Lesion hemisphere, *n* (%)				0.544	0.797
right	13 (56.5)	11 (45.8)	12 (50)		
left	10 (43.5)	13 (54.2)	12 (50)		

**Table 2 ijerph-20-00930-t002:** HAMD-17 scores in A, B, C ground at T_0_, T_1_, T_2_ in terms of mean ± standard error (SE).

Group	Samples	Pre-Test	Post-Test	8-Week Follow-Up	*F*	*p*
Group A	23	19.04 ± 2.16	13.96 ± 2.99 ^a^	11.96 ± 3.24 ^a^	80.501	<0.001
Group B	24	19.13 ± 3.05	15.63 ± 2.02 ^ab^	13.96 ± 2.14 ^ab^	43.833	<0.001
Group C	24	20.25 ± 1.39	17.38 ± 1.41 ^abc^	17.54 ± 2.25 ^abc^	19.082	<0.001
*F*		2.032	13.602	29.421		
*p*		0.139	<0.001	<0.001		

Compared with before treatment, ^a^
*p* < 0.05; compared with group A, ^b^
*p* < 0.05; compared with group C, ^c^
*p* < 0.05. Comparisons between groups after intervention and at the 8-week follow-up were corrected for TO scores using MoCA, MBI, and PSQI.

**Table 3 ijerph-20-00930-t003:** MMSE scores in A, B, C ground at T_0_, T_1_, T_2_ in terms of mean ± standard error (SE).

Group	Samples	Pre-Test	Post-Test	8-Week Follow-Up	*F*	*p*
Group A	23	21.09 ± 4.33	26.48 ± 1.27 ^a^	27.00 ± 1.28 ^a^	30.810	<0.001
Group B	24	20.83 ± 5.59	25.46 ± 2.90 ^a^	25.04 ± 1.63 ^ab^	28.455	<0.001
Group C	24	20.04 ± 5.83	22.50 ± 3.86 ^abc^	21.54 ± 3.43 ^bc^	12.739	<0.001
*F*		0.250	12.021	33.436		
*p*		0.779	<0.001	<0.001		

Compared with before treatment, ^a^
*p* < 0.05; compared with group A, ^b^
*p* < 0.05; compared with group C, ^c^
*p* < 0.05.

**Table 4 ijerph-20-00930-t004:** MBI scores in A, B, C ground at T_0_, T_1_, T_2_ in terms of mean ± standard error (SE).

Group	Samples	Pre-Test	Post-Test	8-Week Follow-Up	*F*	*p*
Group A	23	46.52 ± 19.45	75.43 ± 9.40 ^a^	81.96 ± 9.50 ^a^	97.753	<0.001
Group B	24	45.00 ± 18.00	66.46 ± 9.03 ^ab^	75.00 ± 7.37 ^ab^	87.982	<0.001
Group C	24	43.75 ± 12.87	65.83 ± 8.43 ^ab^	71.46 ± 6.83 ^ab^	63.619	<0.001
*F*		0.157	8.395	10.528		
*p*		0.855	0.001	<0.001		

Compared with before treatment, ^a^
*p* < 0.05; compared with group A, ^b^
*p* < 0.05.

**Table 5 ijerph-20-00930-t005:** PSQI scores in A, B, C ground at T_0_, T_1_, T_2_ in terms of mean ± standard error (SE).

Group	Samples	Samples	Pre-Test	8-Week Follow-Up	*F*	*p*
Group A	23	10.87 ± 1.87	7.43 ± 1.70 ^a^	4.35 ± 1.19 ^a^	104.631	<0.001
Group B	24	11.79 ± 2.54	9.08 ± 2.62 ^ab^	6.88 ± 2.85 ^ab^	64.902	<0.001
Group C	24	10.54 ± 2.34	8.67 ± 2.20 ^a^	8.75 ± 2.94 ^abc^	25.688	<0.001
*F*		1.952	3.508	18.688		
*p*		0.150	0.035	<0.001		

Compared with before treatment, ^a^
*p* < 0.05; compared with group A, ^b^
*p* < 0.05; compared with group C, ^c^
*p* < 0.05.

## Data Availability

Not applicable.

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
