# Peer review of "Effectiveness Evaluation of Repetitive Transcranial Magnetic Stimulation Therapy Combined with Mindfulness-Based Stress Reduction for People with Post-Stroke Depression: A Randomized Controlled Trial"

_ijerph, 2023, doi:10.3390/ijerph20020930_

Round 1
Reviewer 1 Report
Review
This study evaluated the effectiveness of a combined intervention based on rTMS and MBSR for the physical and mental state of PSD patients.
General comments
This is an interesting study that could make a valuable contribution to the literature. The manuscript would benefit from significant editing for typos, English language use and word choice. In particular, the introduction and discussion sections would benefit from editing for clarity. Both sections could likely be shortened to make the article more reader friendly.
Abstract
The results section could benefit from editing for clarity. Does, “A total of 71 patients completed the questionnaire” mean 71 subjects were enrolled? Did all complete the study? Please clarify, “The repeated-measure ANOVA showed a significant improvement of all variables in rTMS-MBSR than sham rTMS-MBSR and sham rTMS combined with general psychological care(P<0.05).” Should “than” be “compared to?”
Introduction
Line 57. “…is often a vital stimulation target…”. Is vital the correct word?
Lines 68 -70. How does this sentence tie in to the one above about mindfulness?
Lines 74 – 77. What does, “Bermudo et al. 74 will explore…” mean? I this a future study? If so why is it relevant if there are no results. “Unfortunately, MBSR is not widely studied to improve negative affect in clinical for PSD patients.” Do you mean “has not?” … affect in clinical for PSD” needs to be reworded.
Line 90. Is “profound” the correct word?
Materials and methods
What were the recruitment methods? Were subjects compensated for participation.
Were subjects blinded at all to the treatments they received?
Section 2.2. What were the ranges of time post-stroke and duration of depression for subjects? Were these factors considered in inclusion/exclusion criteria or as confounding variables?
Line 117. Consider rewording “lack of cooperation.” How was “severe agitation” designed.
Section 2.4. Clarify that the intervention the MBSR/rTMS, but that participants also received routine standard of care medical treatment.
Lines 153 and 154. This sentence needs to be reworded.
Line 155. Was the instructor certified to provide MBSR?
Section 2.4.3. Provide more information. How many sessions & how frequent? Were the sessions timed to match the MBSR sessions in frequency and duration. Is this a control intervention or treatment as usual?
Discussion
Line 298. Is placebo the correct word?
Lines 300 – 301. What does, “matched the 300 treatment's efficacy” mean?
Lines 304 – 305. “Our results further support the theory of A cognitive perspective for Depression and Monitoring and Acceptance Theory (MAT).” What are these theories and how specifically does the study support them.
Lines 306 -308. Explain how the study supports the use of MBSR alone for PSD.
Line 312. What is “mindfulness decompression therapy?
Reviewer 2 Report
1. This is a clinical RCT, so the authors are required to provide the consort diagram.
2. In 2.5. Data Collection. Authors are required to mention any dropout in post- and 8-week follow-ups. It seems no such issue. If there is no dropout, that's fine. Otherwise, you need to compare results between ITT and pre-protocol, I hope no need.
3. Authors are required to provide the response rate in the abstract and CONSORT diagram.
4. 2.3. Sample size estimation. It was estimated based on one-way ANOVA, but your design is RM ANOVA / GLM, so you need to cross-check this part again. It seems you only focus on the ES (8 weeks vs. baseline) among 3 groups (between-effect), so interaction effect (between group x time) you won't consider, so why do you collect post-outcome?
5. One of your presentations uses correction on HAMD-17. I think you have to use partial correction, controlled by other outcomes.
Round 2
Reviewer 1 Report
The authors have adequately addressed my concerns. I have no further comments.